# Ganoderic Acid A and Its Amide Derivatives as Potential Anti-Cancer Agents by Regulating the p53-MDM2 Pathway: Synthesis and Biological Evaluation

**DOI:** 10.3390/molecules28052374

**Published:** 2023-03-04

**Authors:** Yi Jia, Yan Li, Hai Shang, Yun Luo, Yu Tian

**Affiliations:** Institute of Medicinal Plant Development, Chinese Academy of Medical Sciences & Peking Union Medical College, Beijing 100193, China

**Keywords:** Ganoderic acid A, derivatives, p53, MDM2, tumor, mechanism of action

## Abstract

The mechanisms of action of natural products and the identification of their targets have long been a research hotspot. Ganoderic acid A (GAA) is the earliest and most abundant triterpenoids discovered in *Ganoderma lucidum*. The multi-therapeutic potential of GAA, in particular its anti-tumor activity, has been extensively studied. However, the unknown targets and associated pathways of GAA, together with its low activity, limit in-depth research compared to other small molecule anti-cancer drugs. In this study, GAA was modified at the carboxyl group to synthesize a series of amide compounds, and the in vitro anti-tumor activities of the derivatives were investigated. Finally, compound **A2** was selected to study its mechanism of action because of its high activity in three different types of tumor cell lines and low toxicity to normal cells. The results showed that **A2** could induce apoptosis by regulating the p53 signaling pathway and may be involved in inhibiting the interaction of MDM2 and p53 by binding to MDM2 (K_D_ = 1.68 µM). This study provides some inspiration for the research into the anti-tumor targets and mechanisms of GAA and its derivatives, as well as for the discovery of active candidates based on this series.

## 1. Introduction

*Ganoderma lucidum* is the dry fruiting body of *Ganoderma lucidum* Karst and *Ganoderma sinensis*, which belong to the genus Ganoderma of the family Polyporaceae. The chemical composition of *Ganoderma lucidum* is complex. There are currently about 400 known compounds, the majority of which are triterpenoids, polysaccharides, nucleosides, sterols, and other compounds, of which more than 300 are triterpenoids. Ganoderic acid A (GAA, Figure 1), one of the most prominent and highly concentrated triterpenes from *Ganoderma lucidum*, exhibits a variety of biological properties, including anti-tumor [1], anti-inflammatory [2,3,4], anti-depressant [5,6], neuroprotection [7,8], anti-fibrosis [9], liver protection [10,11], improvement of glucose and lipid metabolism and myocardial protection [12,13,14], etc., which can be used as a potential resource for drug development. The anti-tumor activity is one of the earliest discovered activities of GAA, which has received the most attention since then [1]. Many researchers have investigated the anti-tumor activities of *Ganoderma lucidum* triterpenoids and predicted their anti-tumor pathway. Studies have shown that GAA can inhibit tumor growth through a variety of signaling pathways. For example, GAA has good cytotoxicity on human glioblastoma by inducing apoptosis, autophagy and inhibiting PI3K/AKT signaling pathways [15]; it can inhibit the expression of *KDR* mRNA and protein, induce apoptosis of human glioma cell U251 cell, and inhibit its proliferation and invasion [16]. However, there is no relevant literature that clearly indicates the possible anti-tumor target of GAA.

Cancer and the MDM2-p53 signaling pathway are closely related. p53 is a tumor suppressor gene. When cells are damaged by a variety of causes such as DNA damage, ribosomal stress, the expression of the p53 protein is activated to repair damaged cells or to directly induce apoptosis if the DNA damage is already too severe. p53 is essential for a number of processes that occur throughout life, including DNA damage repair, cell cycle arrest, metabolism, senescence, and apoptosis [17]. If too much p53 protein is produced during certain physiological processes, cell function is impaired or the tendency to form tumors is increased. Therefore, the expression of murine double minute 2 (MDM2) protein in the downstream signaling pathway will increase when p53 protein accumulates in normal cells. To achieve the balance and stability of p53 protein levels in cells, MDM2 can interact with the transcriptional activation domain of the p53 to form the p53-MDM2 complex, which suppresses the transcriptional activity of p53. When a cell is stressed, MDM2 expression decreases, p53 expression increases, and the increase in p53 induces MDM2 expression at the transcriptional level, creating a negative feedback regulatory loop (Figure 2) [18]. MDM2 can also act as an E3 ubiquitin ligase, targeting p53 protein and inducing its ubiquitination and degradation to maintain low levels of p53 protein [19]. p53 has long been an intriguing cancer target [20]. Individuals carrying certain inherited loss-of-function mutations in p53 have a 50% chance of developing cancer by the age of 30 and a 90% chance of developing cancer by the age of 70 [21]. Mice knocked out of p53 quickly develop tumors. Up to 50% of cancers have mutations in both copies of p53 [22]. Drugs that can reactivate the tumor suppressing ability of p53 may therefore have a powerful anti-cancer effect. However, it is more difficult to activate proteins than to inhibit them, so the interaction of MDM2 with p53 provides an opportunity to activate p53 by inhibiting the interaction of MDM2 to exert anti-tumor effects.

After summarizing the relevant literature, we discovered that GAA may interact with the p53-MDM2 pathway. For example, Xu Bin et al. found that GAA inhibited LNCaP in a concentration-dependent manner. Real-time experiments showed that GAA promoted the apoptosis-promoting genes bad and p53 [23]. Tang Wen et al. found that 95-D cells expressing wild-type p53 protein were 3.3 times more sensitive to ganoderic acid T than H1299 cells that did not express p53 protein [24]. Other studies suggest that GAA and sterols with similar structures may have some affinity for the MDM2. Froufe et al. found that some *Ganoderma lucidum* triterpenoids have potential affinities with MDM2 protein through virtual screening prediction, including ganoderic acid A (K_i_ = 147 nM) and ganoderic acid F (K_i_ = 212 nM) [25]. Staszczak et al. summarized the role of secondary metabolites in fungi on the ubiquitin–protesome system, in which sterols have certain interactions with MDM2, indicating that such structure has advantages in interactions with MDM2 [26]. All these results suggest that GAA is likely to be related to the p53-MDM2 pathway. However, considering the low anti-tumor effect of GAA and the size of the pocket of MDM2, and there is no relevant literature highlighting the anti-cancer activities of synthetic GAA derivatives on potential MDM2-p53 interaction inhibitions, we decided to simply modify the structure of GAA at the carboxyl group to improve its anti-tumor activity and reduce possible pharmacokinetic problems caused by the free carboxyl group, and investigated the effects of the different GAA amide derivatives on the MDM2-p53 pathway.

In this study, GAA was modified to determine the in vitro anti-tumor activity of these derivatives on different tumor cell lines, and compound **A2** (Figure 1), which has good activity in different cell lines and low toxicity to normal cells, was selected to investigate the relevant mechanism. First, we investigated the effect of **A2** on cell apoptosis and the expression of proteins related to the MDM2-p53 pathway by flow cytometry and Western blot experiment. Next, in silico target fishing and molecular docking was performed to investigate the binding potential of **A2** and MDM2. We then used a surface plasmon resonance (SPR) experiment to show that **A2** has a certain binding affinity with MDM2 in vitro. It was speculated that **A2** might play a role in increasing p53 protein levels by binding to MDM2 to inhibit the interaction of MDM2 and p53. This work is valuable in further demonstrating the potential of GAA and its amide derivatives as MDM2-p53 binding inhibitors and in developing candidates with anti-tumor activity.

## 2. Results and Discussion

### 2.1. Chemistry

Based on the structure of GAA, we retained its core structure of tetracyclic triterpenoids, and introduced a series of amino groups to modify GAA at the carboxyl site. As shown in Figure 1, GAA was treated with amino compounds, 2-(1*H*-benzotriazole-1-yl)-1,1,3,3-tetramethyluronium tetrafluoroborate (TBTU), and *N*,*N*-diisopropylethylamine (DIPEA) to obtain GAA derivatives [27,28]. **A1**–**A12** refers to the amide derivatives formed with fatty amine, aniline, benzylamine, phenylethylamine and other different types of primary amine compound. **A13**–**A15** refers to the derivatives formed with piperazine compounds.

Except for the low yield of substituted aniline, the yields of the other compounds are 70~98%, which is easy to obtain. See Methods and Materials for the detailed synthesis and purification methods of all compounds. After being substituted by different amine fragments, the hydrogen signal of amide bond appears at 7–5 ppm. The methylene peak of amine fragments is mostly distributed at 4.5–3 ppm. The chemical shift of hydrogen signal in GAA itself does not change very much. All new compounds were identified by ^1^H-NMR, ^13^C-APT and HRMS spectroscopy. The corresponding spectra are presented in the Appendix A.

### 2.2. In Vitro Anti-Proliferation Activity

#### 2.2.1. The Anti-Proliferation Activity on MCF-7

MCF-7 is a commonly used tumor cell line. Previous studies have shown that GAA has some anti-tumor activity against MCF-7, and there is a high expression of MDM2-p53 in MCF-7. The anti-tumor activities of GAA derivatives on MCF-7 were tested for 48 h, and the results were shown in Table 1 and Figure 3. The results showed that compounds **A2**, **A6**, **A7**, **A8**, **A9**, **A15** had significant anti-proliferation activities on MCF-7 cell line compared with GAA. Among all derivatives, **A6** has the strongest anti-proliferation effect, and its inhibition rate of MCF-7 at 50 µM can reach 63.64%.

Overall, among aliphatic amines, anilines, benzylamines, phenylethylamines, and (hetero) cyclic amines, benzylamine derivatives (**A6**, **A7**, **A8**) were significantly more potent than other substituted compounds. Among the aliphatic amines (**A1**, **A2**), the chain length of six carbon atoms is better than that of four carbon atoms. Among benzylamine compounds, the activities of electron withdrawing groups on benzene ring (**A6**, **A7**, **A8**) (cell viability at 50 μM less than 50%) are better than that of electron donating group (**A5**), and 3,5-diCl double substitution is better than 4-Cl single substitution, indicating that the position and amount of electron withdrawing groups can affect the activities of GAA derivatives. Compared with anilines (**A3**, **A4**), benzylamines (**A6**, **A7**, **A8**) and phenylethylamines (**A10**, **A11**, **A12**) substituted compounds, the activities of benzylamines are better, which also indicated that the chain length of substituents may affect their activities. At the same time, the introduction of the common anti-tumor fragment indene can improve the anti-proliferative activity of GAA (**A9**). The introduction of *N*-methyl or *N*-ethyl piperazine with strong hydrophilicity can’t improve the anti-tumor activity of GAA, but *N*-phenyl piperazine can improve activity (**A13**, **A14**, **A15**), indicating that the anti-proliferation activities of GAA derivatives may have certain requirements for hydrophobicity.

#### 2.2.2. The Anti-Proliferation Activity on SJSA-1, HepG2 and HK2

To investigate the selectivity of these derivatives towards different tumor cell lines, we also selected HepG2 and osteosarcoma cell line SJSA-1 cells to evaluate the anti-proliferation activity of the derivatives. The results were shown in Table 2 and Figure 3. The results showed that the inhibitory effect of this series of derivatives on HepG2 was overall better than that on MCF-7 on the whole. Except for compounds **A2** and **A11**, the effects of other compounds on SJSA-1 were not strong. In HepG2 cell line, compounds **A2**, **A7**, **A8** and **A9** still have potent anti-proliferation activity, whereas **A6** and **A15**, which were better in MCF-7, have weaker anti-proliferation effect on HepG2. However, **A12** had strong selectivity on HepG2, and the inhibition rate of this cell below 50 µM can reach 74.37%. In SJSA-1 cell line, compound **A2** still showed potent inhibition, whereas **A11** showed some selectivity for SJSA-1, and it was found that GAA had better anti-tumor activity for SJSA-1 than for HepG2 and MCF-7.

### 2.3. ***A2*** Induces Apoptosis in SJSA-1 Cells

We next examined the effect of **A2** (24 h incubation, at concentrations of 12.5, 25, 50 µM) on the SJSA-1 which **A2** showed the highest anti-proliferation potency among all the cell lines. Cells were stained with Annexin V-FITC and propidium iodide. The results are shown in the Figure 4. The results showed that different concentrations of **A2** could induce different degrees of apoptosis in SJSA-1 cells. At low concentrations, the proportion of cells undergoing early apoptosis increased slightly from 11.6% (12.5 µM) to 12.3% (25 µM). However, the proportion of apoptosis cells increased significantly at 50 µM (18.7%), while the proportion of late apoptosis remained essentially unchanged with increasing of concentration. The results indicated that **A2** can induce cell apoptosis in a dose-dependent manner.

### 2.4. ***A2*** Effects MDM2-P53 Signaling Pathway

#### 2.4.1. **A2** Effects the Expression of p53 Protein, MDM2 and Bcl-2/Bax

In the introduction section, we introduced that the concept that the MDM2-p53 pathway can induce cell apoptosis by up-regulating the expression of p53 protein to inhibit the proliferation of tumor cells, and by blocking the interaction of MDM2 and p53, the activation of p53 results in transcription of *MDM2* mRNA, leading to robust MDM2 protein accumulation [29,30,31]. In order to verify the effect of **A2** on this pathway, we examined the effect of **A2** on the protein level of MDM2, p53 protein and Bcl-2/Bax related to apoptosis, as shown in Figure 5. The results showed that after treatment of MCF-7 cells with **A2** for 24 h, both MDM2 and p53 protein showed an increasing trend at 50 µM. The level of Bcl-2/Bax decreased which was consistent with the apoptosis of MCF-7 cells induced by **A2**. We also investigated the effect of **A2** on the SJSA-1 cell line which overexpresses MDM2. Compared with MCF-7, the expression of MDM2 and p53 protein in this cell line increased in a dose-dependent manner which may be the reason for the best anti-proliferation effect on SJSA-1 among all three cell lines. This experiment demonstrated that **A2** can affect the MDM2-p53 pathway to induces apoptosis.

#### 2.4.2. GAA and **A2** Have In Vitro Binding Affinity with MDM2

In order to speculate whether **A2** effects the p53-MDM2 pathway by binding with MDM2 to inhibit the interaction between MDM2 and p53, we carried out in silico and in vitro binding experiments. First, we performed computer simulation to conduct target fishing of GAA and found that MDM2 interacts with GAA in silico (FitValue 0.79). We used the S-value to evaluate the binding degree of the compound and MDM2 in the molecular docking experiments. The higher the absolute value of this number, the stronger the binding force. Molecular docking (see Figure 6) revealed that the hydroxyl-H of GAA interacts with Met58 in MDM2 (S-value: −6.49). When **A2** was docked to MDM2, it was found that, the core of **A2** was in the opposite direction compared to GAA. In addition to the interaction with Met58 similar to GAA, the *n*-hexyl is well anchored in the hydrophobic pocket and the methylene has some hydrophobic interaction with His92 (S-value: −7.22). To verify whether GAA and **A2** have a certain binding ability with MDM2 in vitro, we used surface plasmon resonance (SPR) experiment to investigate the interaction between GAA, **A2** and MDM2 (see Figure 7). The K_D_ of GAA and MDM2 is 12.73 µM, indicating that they do have some affinity. At the same time, **A2** which has a stronger anti-proliferation activity has a stronger binding affinity with MDM2 than GAA, with a K_D_ of 1.68 µM. These results demonstrated that **A2** can affect the MDM2-p53 pathway to induces apoptosis probably by inhibiting the interaction of MDM2 and p53.

We then investigated the effects of compounds with significantly higher activity than GAA in three cell lines on HK2, which is a normal cell line used to assess cytotoxicity. The results are shown in Table 2 and Figure 3. The results showed that at high concentration, benzylamine compounds **A6**, **A7** and **A9** with anti-tumor fragments had some toxicity to HK2 cells, whereas the other compounds with stronger activity had lower cytotoxicity to HK2 cells. To sum up, this series of GAA derivatives showed some selectivity in different cell lines, and have the potential to be developed as various tumor inhibitors. Given the strong anti-proliferation effect of derivative **A2** in various cell lines and its low effect on normal cells, **A2** was selected to investigate its anti-proliferation mechanism.

## 3. Materials and Methods

### 3.1. Chemistry

Unless otherwise stated, all reagents and solvents were obtained from commercial sources were used without further purification. GAA were purchased from Biopurify (Chengdu, China). Flash column chromatography was performed on Biotage Isolera Four (Sweden). ^1^H NMR and ^13^C-APT spectra were recorded on a Bruker AvanceIII 600MHz spectrometer (Germany). HRMS was performed on a Thermo Fisher LTQ Orbitrap XL (United States).

#### 3.1.1. Synthesis of (*n*-butyl)-(7β,15α,25R)-7,15-Dihydroxy-3,11,23-Trioxolanost-8-en-26-oic Amide (**A1**)

To a solution of GAA (1eq., 50 mg, 0.01 mmol) in DCM (5 mL), *n*-butylamine (2 eq.), TBTU (1.5 eq.) and DIPEA (1.5 eq.) were added. The resulting reaction mixture was stirred at room temperature for 1 h and monitored by TLC. Upon completion, the reaction was then quenched with water and extracted with DCM. The organic layer was washed twice with water, dried over anhydrous sodium sulfate, filtered and concentrated under reduced pressure. The crude material was purified by column chromatography using dichloromethane and methanol (10:1, *v*/*v*) as mobile phase to obtain target molecule as white powder (yield 80.6%). mp: 120.7–121.5 °C. ^1^H-NMR (600 MHz, CDCl_3_) *δ*: 5.81 (t, *J* = 5.6 Hz, 1H, CO*N*H), 4.80–4.77 (m, 1H, H-15), 4.63–4.60 (m, 1H, H-7), 4.16–4.15 (m, 1H, OH-15), 3.55–3.52 (m, 1H, OH-7), 3.22–3.19 (m, 2H, CO*N*HCH_2_), 2.90–2.79 (m, 2H, H-24a, H-1b), 2.78–2.68 (m, 2H, H-12a, H-25), 2.54–2.45 (m, 3H, H-22a, H-12b, H-24b), 2.44–2.37 (m, 2H, H-2), 2.26–2.19 (m, 1H, H-22b), 2.07–2.01 (m, 1H, H-6a), 2.01–1.95 (m,1H, H-20), 1.83–1.78 (m, 3H, H-17, H-16), 1.74–1.66 (m, 2H, H-5, H-6b), 1.52–1.42 (m, 3H, H-1a, CH_2_), 1.37–1.30 (m, 2H, CH_2_), 1.27 (s, 3H, CH_3_), 1.25 (s, 3H, CH_3_), 1.14 (d, *J* = 7.0 Hz, 3H, CH_3_), 1.12 (s, 3H, CH_3_), 1.10 (s, 3H, CH_3_), 0.99 (s, 3H, CH_3_), 0.91 (t, *J* = 7.3 Hz, 3H, CH_2_CH_3_), 0.87 (d, *J* = 6.4 Hz, 3H, CH_3_). ^13^C-APT (150 MHz, CDCl_3_) *δ*: 217.3, 209.7, 199.6, 175.5, 159.3, 140.1, 72.3, 68.8, 53.9, 51.7, 49.8, 48.7, 48.1, 47.2, 46.7, 46.6, 39.3, 37.9, 36.3, 36.0, 35.5, 34.3, 32.7, 31.6, 28.9, 27.3, 20.7, 20.0, 19.6, 19.5, 19.4, 18.0, 17.3, 13.8. HRMS calculated for C_34_H_53_NO_6_Na [M + Na]^+^ *m/z* 594.3765, found 594.3746.

#### 3.1.2. Synthesis of (*n*-hexyl)-(7β,15α,25R)-7,15-Dihydroxy-3,11,23-Trioxolanost-8-en-26-oic Amide (**A2**)

The title compound was obtained from 1-hexanamine following similar synthesis procedure of **A1** (white powder, yield 65.6%). mp: 129.9–131.3 °C. ^1^H-NMR (600 MHz, CDCl_3_) *δ*: 5.78 (t, *J* = 5.6 Hz, 1H, CO*N*H), 4.79–4.77 (m, 1H, H-15), 4.64–4.60 (m, 1H, H-7), 4.06–3.96 (m, 1H, OH), 3.50–3.35 (m, 1H, OH), 3.21–3.17 (m, 2H, CO*N*HCH_2_), 2.91–2.80 (m, 2H, H-24a, H-1b), 2.78–2.68 (m, 2H, H-12a, H-25), 2.54–2.45 (m, 3H, H-22a, H-12b, H-24b), 2.44–2.37 (m, 2H, H-2), 2.26–2.19 (m, 1H, H-22b), 2.09–2.02 (m, 1H, H-6a), 2.01–1.95 (m,1H, H-20), 1.84–1.76 (m, 3H, H-17, H-16), 1.74–1.64 (m, 2H, H-5, H-6b), 1.51–1.43 (m, 3H, H-1a, CH_2_), 1.35–1.23 (m, 12H, CH_2_ × 3, CH_3_ × 2), 1.25 (s, 3H, CH_3_), 1.14 (d, *J* = 7.1 Hz, 3H, CH_3_), 1.12 (s, 3H, CH_3_), 1.10 (s, 3H, CH_3_), 0.99 (s, 3H, CH_3_), 0.89−0.86 (m, 6H, 2×CH_3_). ^13^C-APT (150 MHz, CDCl_3_) *δ:* 217.2, 209.7, 199.6, 175.5, 159.3, 140.2, 72.3, 68.8, 54.0, 51.7, 49.9, 48.7, 48.1, 47.2, 46.8, 46.6, 39.6, 38.0, 36.4, 36.0, 35.5, 34.3, 32.8, 31.5, 29.5, 29.0, 27.4, 26.5, 22.6, 20.7, 19.6, 19.5, 19.4, 18.0, 17.3, 14.1. HRMS calculated for C_36_H_57_NO_6_Na [M + Na]^+^ *m/z* 622.4078, found 622.4070.

#### 3.1.3. Synthesis of (4-Methylphenyl)-(7β,15α,25R)-7,15-Dihydroxy-3,11,23-Trioxolanost-8-en-26-oic Amide (**A3**)

The title compound was obtained from *p*-toluidine following similar synthesis procedure of **A1** (white powder, yield 30.3%). mp: 174.3–175.5 °C. ^1^H-NMR (600 MHz, CDCl_3_) *δ*: 7.76 (s, 1H, CO*N*H), 7.29 (d, *J* = 8.2 Hz, 2H, Ph-2, 6-H), 7.02 (d, *J* = 8.2 Hz, 2H, Ph-2, 6-H), 4.66–4.64 (m, 1H, H-15), 4.51–4.50 (m, 1H, H-7), 4.07–4.03 (m, 1H, OH), 3.46-3.32 (m, 1H, OH), 2.92–2.80 (m, 2H, H-24a, H-1b), 2.79–2.72 (m, 1H, H-25), 2.67–2.62 (d, *J* = 16.1 Hz, H-12a), 2.44–2.37 (m, 4H, H-22a, H-12b, H-24b, H-2a), 2.36–2.30 (m, 1H, H-2b), 2.23 (s, 3H, Ph-CH_3_), 2.21–2.14 (m, 1H, H-22b), 1.96–1.92 (m, 1H, H-6a), 1.92–1.87 (m,1H, H-20), 1.76-1.67 (m, 3H, H-17, H-16), 1.62–1.58 (m, 2H, H-5, H-6b), 1.42–1.34 (m, 1H, H-1a), 1.18 (s, 3H, CH_3_), 1.16 (d, *J* = 6.7 Hz, 3H, CH_3_), 1.15 (s, 3H, CH_3_), 1.03 (s, 3H, CH_3_), 1.01 (s, 3H, CH_3_), 0.89 (s, 3H, CH_3_), 0.78 (d, *J* = 6.7 Hz, 3H, CH_3_). ^13^C-APT (150 MHz, CDCl_3_) *δ*: 216.4, 209.2, 198.7, 172.9, 158.3, 139.0, 134.2, 133.0, 128.4, 119.0, 71.2, 67.7, 52.9, 50.6, 48.8, 47.6 47.0, 46.3, 45.7, 45.6, 36.9, 35.7, 35.1, 34.4, 33.2, 31.8, 27.8, 26.3, 19.9, 19.6, 18.6, 18.4, 18.4, 16.9, 16.2. HRMS calculated for C_37_H_51_NO_6_Na [M + Na]^+^ *m/z* 628.3609, found 628.3594.

#### 3.1.4. Synthesis of (4-Chlorophenyl)-(7β,15α,25R)-7,15-Dhydroxy-3,11,23-Trioxolanost-8-en-26-oic Amide (**A4**)

The title compound was obtained from *p*-chloroaniline following similar synthesis procedure of **A1** (white powder, yield 26.3%). mp: 184.9–185.3 °C. ^1^H-NMR (600 MHz, CDCl_3_) *δ*: 7.78 (s, 1H, CO*N*H), 7.37 (d, *J* = 8.6 Hz, 2H, Ph-2, 6-H), 7.17 (d, *J* = 8.7 Hz, 2H, Ph-2, 6-H), 4.69–4.66 (m, 1H, H-15), 4.55–4.52 (m, 1H, H-7), 2.93–2.80 (m, 2H, H-24a, H-1b), 2.79–2.71 (m, 1H, H-25), 2.69–2.63 (d, *J* = 16.1 Hz, H-12a), 2.47–2.38 (m, 4H, H-22a, H-12b, H-24b, H-2a), 2.36–2.30 (m, 1H, H-2b), 2.21–2.14 (m, 1H, H-22b), 1.99–1.94 (m, 1H, H-6a), 1.94–1.88 (m,1H, H-20), 1.74–1.69 (m, 3H, H-17, H-16), 1.64–1.57(m, 2H, H-5, H-6b), 1.42–1.35(m, 1H, H-1a), 1.19–1.55 (m, 9H, 3 × CH_3_), 1.04 (s, 3H, CH_3_), 1.01 (s, 3H, CH_3_), 0.89 (s, 3H, CH_3_), 0.78 (d, *J* = 6.2 Hz, 3H, CH_3_). ^13^C-APT (150 MHz, CDCl_3_) *δ*: 216.3, 209.3, 198.5, 173.0, 158.1, 139.1, 135.5, 128.1, 127.9, 120.0, 71.3, 67.8, 52.8, 50.6, 48.7, 47.6, 47.0, 46.4, 45.7, 45.6, 36.9, 35.6, 35.2, 34.4, 33.2, 31.8, 27.9, 26.3, 19.6, 18.6, 18.4, 18.3, 16.8, 16.2. HRMS calculated for C_36_H_48_ClNO_6_Na [M + Na] ^+^ *m/z* 648.3062, found 648.3049.

#### 3.1.5. Synthesis of (4-methylbenzyl)-(7β,15α,25R)-7,15-Dihydroxy-3,11,23-Trioxolanost-8-en-26-oic Amide (**A5**)

The title compound was obtained from 4-methylphenyl following similar synthesis procedure of **A1** (white powder, yield 91.9%). mp: 208.7–209.0 °C. ^1^H-NMR (600 MHz, CDCl_3_) *δ*: 7.06 (m, 4H, Ph-H), 6.09 (t, *J* = 7.2 Hz, 1H, CO*N*H), 4.70–4.67 (m, 1H, H-15), 4.53–4.51 (m, 1H, H-7), 4.29–4.28 (m, 2H, CO*N*HCH_2_), 4.21–4.24 (m, 1H, OH), 3.64–3.45 (m, 1H, OH), 2.87–2.79 (m, 1H, H-1b), 2.78–2.73 (m, 1H, H-24a), 2.72-2.64 (m, 2H, H-25, H-12a), 2.45–2.30 (m, 5H, H-22a, H-12b, H-24b, H-2), 2.26 (s, 3H, Ph-CH_3_), 2.20–2.12 (m, 1H, H-22b), 1.97–1.93 (m, 1H, H-6a), 1.92–1.87 (m,1H, H-20), 1.76–1.69 (m, 3H, H-17, H-16), 1.64–1.56 (m, 2H, H-5, H-6b), 1.43–1.34 (m, 1H, H-1a), 1.19 (s, 3H, CH_3_), 1.18 (s, 3H, CH_3_), 1.10 (d, *J* = 7.11 Hz, 3H, CH_3_), 1.03 (s, 3H, CH_3_), 1.01 (s, 3H, CH_3_), 0.90 (s, 3H, CH_3_), 0.78 (d, *J* = 6.4 Hz, 3H, CH_3_). ^13^C-APT (150 MHz, CDCl_3_) *δ*: 216.4, 208.6, 198.7, 174.6, 158.5, 139.0, 136.1, 134.0, 128.3, 126.5, 71.2, 67.7, 52.9, 50.6, 48.8, 47.6, 47.0, 46.1, 45.7, 45.6, 42.3, 36.9, 35.2, 34.9, 34.5, 33.3, 31.7, 27.8, 26.3, 20.1, 19.6, 18.6, 18.5, 18.4, 17.0, 16.2. HRMS calculated for C_38_H_53_NO_6_Na [M + Na]^+^ *m/z* 642.3765, found 642.3752.

#### 3.1.6. Synthesis of (4-Fluorobenzyl)-(7β,15α,25R)-7,15-Dihydroxy-3,11,23-Trioxolanost-8-en-26-oic Amide (**A6**)

The title compound was obtained from *p*-fluorobenzylamine following similar synthesis procedure of **A1** (white powder, yield 90.2%). mp: 178.4–179.3 °C. ^1^H-NMR (600 MHz, CDCl_3_) *δ*: 7.16 (m, 2H, Ph-2, 6-H), 6.93 (m, 2H, Ph-3, 5-H), 6.18 (m, 1H, CO*N*H), 4.70–4.67 (m, 1H, H-15), 4.53–4.52 (m, 1H, H-7), 4.34–4.26 (m, 2H, CO*N*HCH_2_), 4.12–3.98 (m, 1H, OH), 3.56–3.37 (m, 1H, OH), 2.88–2.80 (m, 1H, H-1b), 2.78–2.64 (m, 3H, H-24a, H-12a, H-25), 2.47–2.30 (m, 5H, H-22a, H-12b, H-24b, H-2), 2.18–2.11 (m, 1H, H-22b), 1.99–1.93 (m, 1H, H-6a), 1.92–1.87 (m,1H, H-20), 1.75–1.69 (m, 3H, H-17, H-16), 1.64–1.58 (m, 2H, H-5, H-6b), 1.43–1.35 (m, 1H, H-1a), 1.19 (s, 3H, CH_3_), 1.17 (s, 3H, CH_3_), 1.10 (d, *J* = 7.4 Hz, 3H, CH_3_), 1.04 (s, 3H, CH_3_), 1.01 (s, 3H, CH_3_), 0.91 (s, 3H, CH_3_), 0.78 (d, *J* = 6.4 Hz, 3H, CH_3_). ^13^C-APT (150 MHz, CDCl_3_) *δ*: 216.3, 208.7, 198.6, 174.7, 161.9, 158.3, 139.1, 132.97, 132.95, 128.18, 128.13, 114.55, 114.41, 71.3, 67.8, 52.9, 50.6, 48.7, 47.6, 47.0, 46.1, 45.7, 45.6, 41.8, 36.9, 35.2, 34.9, 34.4, 33.2, 31.7, 28.7, 27.9, 26.3, 19.6, 18.6, 18.5, 18.4, 17.0, 16.2. HRMS calculated for C_37_H_51_FNO_6_ [M + H]^+^ *m/z* 624.3695, found 624.3686.

#### 3.1.7. Synthesis of (4-Chlorobenzyl)-(7β,15α,25R)-7,15-Dihydroxy-3,11,23-Trioxolanost-8-en-26-oic Amide (**A7**)

The title compound was obtained from *p*-chlorobenzylamine following similar synthesis procedure of **A1** (white powder, yield 94.1%). mp: 188.6–189.4 °C. ^1^H-NMR (600 MHz, CDCl_3_) *δ*: 7.21 (d, *J* = 8.5 Hz, 2H, Ph-2, 6-H), 7.11 (d, *J* = 8.5 Hz, 2H, Ph-3, 5-H), 6.28 (m, 1H, CO*N*H), 4.67 (m, 1H, H-15), 4.52 (m, 1H, H-7), 4.34–4.25 (m, 2H, CO*N*HCH_2_), 4.22–4.04 (m, 1H, OH), 3.74–3.38 (m, 1H, OH), 2.87–2.80 (m, 1H, H-1b), 2.78–2.64 (m, 3H, H-24a, H-12a, H-25), 2.46–2.28 (m, 5H, H-22a, H-12b, H-24b, H-2), 2.18-2.11 (m, 1H, H-22b), 2.00–1.93 (m, 1H, H-6a), 1.93–1.86 (m, 1H, H-20), 1.75–1.67 (m, 3H, H-17, H-16), 1.64–1.58 (m, 2H, H-5, H-6b), 1.44–1.33 (m, 1H, H-1a), 1.19 (s, 3H, CH_3_), 1.16 (s, 3H, CH_3_), 1.10 (d, *J* = 7.0 Hz, 3H, CH_3_), 1.04 (s, 3H, CH_3_), 1.01 (s, 3H, CH_3_), 0.90 (s, 3H, CH_3_), 0.78 (d, *J* = 7.0 Hz, 3H, CH_3_). ^13^C-APT (150 MHz, CDCl_3_) *δ*: 216.4, 208.7, 198.7, 174.8, 158.3, 139.0, 135.8, 132.1, 127.8, 127.7, 71.2, 67.7, 52.9, 50.6, 48.7, 47.6, 47.0, 46.1, 45.7, 45.6, 41.8, 36.9, 35.2, 34.7, 34.4, 33.2, 31.7, 27.8, 26.3, 19.6, 18.6, 18.5, 18.4, 17.0, 16.2. HRMS calculated for C_37_H_50_ClNO_6_Na [M + Na]^+^ *m/z* 662.3219, found 662.3206.

#### 3.1.8. Synthesis of (3,5-Dichlorobenzyl)-(7β,15α,25R)-7,15-Dihydroxy-3,11,23-Trioxolanost-8-en-26-oic Amide (**A8**)

The title compound was obtained from 3,5-dichlorobenzylamine following similar synthesis procedure of **A1** (white powder, yield 96.5%). mp: 182.6–183.8 °C. ^1^H-NMR (600 MHz, CDCl_3_) *δ*: 7.17 (m, 1H, Ph-4-H), 7.08 (m, 2H, Ph-2, 6-H), 6.49 (m, 1H, CO*N*H), 4.67 (m, 1H, H-15), 4.53 (m, 1H, H-7), 4.40–4.19 (m, 2H, CO*N*HCH_2_), 2.89–2.80 (m, 1H, H-1b), 2.79–2.64 (m, 3H, H-24a, H-12a, H-25), 2.46–2.36 (m, 4H, H-22a, H-12b, H-24b, H-2a), 2.34–2.27 (m, 1H, H-2b), 2.19–2.12 (m, 1H, H-22b), 1.97–1.89 (m, 2H, H-6a, H-20), 1.76–1.68 (m, 3H, H-17, H-16), 1.64–1.56 (m, 2H, H-5, H-6b), 1.42–1.33 (m, 1H, H-1a), 1.19 (s, 3H, CH_3_), 1.16 (s, 3H, CH_3_), 1.11 (d, *J* = 6.9 Hz, 3H, CH_3_), 1.04 (s, 3H, CH_3_), 1.01 (s, 3H, CH_3_), 0.90 (s, 3H, CH_3_), 0.77 (d, *J* = 6.2 Hz, 3H, CH_3_). ^13^C-APT (150 MHz, CDCl_3_) *δ*: 208.8, 198.8, 175.1, 158.5, 140.9, 139.0, 134.0, 126.4, 124.7, 71.2, 67.7, 52.9, 50.6, 48.5, 47.6, 47.0, 45.6, 41.3, 37.6, 36.9, 35.1, 34.9, 34.5, 33.2, 31.7, 27.8, 26.3, 19.6, 18.6, 18.5, 18.4, 17.0, 16.2. HRMS calculated for C_37_H_49_Cl_2_NO_6_Na [M + Na]^+^ *m/z* 696.2829, found 696.2818.

#### 3.1.9. Synthesis of (2,3-Dihydro-1*H*-inden-2-yl)-(7β,15α,25R)-7,15-Dihydroxy-3,11,23-Trioxolanost-8-en-26-oic Amide (**A9**)

The title compound was obtained from 2-aminoindane HCl following similar synthesis procedure of **A1** (white powder, yield 94.4%). mp: 199.7–200.3 °C. ^1^H-NMR (600 MHz, CDCl_3_) *δ*: 7.23–7.22 (m, 2H, Ph-H), 7.19–7.17 (m, 2H, Ph-H), 6.10 (d, *J* = 7.8 Hz, 1H, CO*N*H), 4.78 (m, 1H, H-15), 4.68–4.64 (m, 1H, CO*N*HCH), 4.63–4.60 (m, 1H, H-7), 4.20–4.12 (m, 1H, OH), 3.62–3.52 (m, 1H, OH), 3.31–3.27 (m, 2H, CO*N*HCHCH_2_), 2.88–2.82 (m, 2H, CO*N*HCHCH_2_), 2.89–2.72 (m, 5H, H-1b, H-24a, H-12a, H-25, CO*N*HCHCH_2_), 2.69–2.61 (m, 1H, CO*N*HCHCH_2_), 2.53–2.35 (m, 5H, H-22a, H-12b, H-24b, H-2), 2.23–2.19 (m, 1H, H-22b), 2.07–2.01 (m, 1H, H-6a), 2.00–1.95 (m, 1H, H-20), 1.84–1.76 (m, 3H, H-17, H-16), 1.75–1.65 (m, 2H, H-5, H-6b), 1.50–1.42 (m, 1H, H-1a), 1.27 (s, 3H, CH_3_), 1.26 (s, 3H, CH_3_), 1.12–1.11 (d, *J* = 7.3 Hz, 6H, 2 × CH_3_), 1.09 (s, 3H, CH_3_), 0.99 (s, 3H, CH_3_), 0.89 (d, *J* = 6.3 Hz, 3H, CH_3_). ^13^C-APT (150 MHz, CDCl_3_) *δ*: 217.1, 209.5, 199.5, 175.3, 159.0, 140.8, 140.7, 140.2, 126.8, 126.7, 124.8, 124.7, 72.4, 68.8, 53.9, 51.7, 50.5, 49.8, 48.7, 48.1, 47.2, 46.8, 46.6, 43.4, 40.1, 40.0, 37.9, 36.4, 35.9, 35.5, 34.3, 32.7, 29.0, 27.3, 20.7, 19.6, 19.5, 19.4, 17.9, 17.3. HRMS calculated for C_39_H_53_NO_6_Na [M + Na]^+^ *m/z* 654.3765, found 654.3760.

#### 3.1.10. Synthesis of (4-Methylphenethyl)-(7β,15α,25R)-7,15-Dihydroxy-3,11,23-Trioxolanost-8-en-26-oic Amide (**A10**)

The title compound was obtained from 2-(4-methylphenyl) ethanamine following similar synthesis procedure of **A1** (white powder, yield 85.6%). mp: 221.9–223.0 °C. ^1^H-NMR (600 MHz, CDCl_3_) *δ*: 7.12 (d, *J* = 8.0 Hz, 2H, Ph-H), 7.09 (d, *J* = 8.0 Hz, 2H, Ph-H), 5.82–5.77 (m, 1H, CO*N*H), 4.80–4.76 (m, 1H, H-15), 4.63–4.59 (m, 1H, H-7), 4.18–4.12 (m, 1H, OH-7), 3.55–3.39 (m, 3H, CO*N*HCH_2_, OH-15), 2.88–2.80 (m, 2H, H-1b, H-24a), 2.78–2.64 (m, 4H, H-12a, H-25, CH_2_Ph), 2.52–2.46 (m, 3H, H-22a, H-12b, H-24b), 2.42–2.36 (m, 2H, H-2), 2.33 (s, 1H, Ph-CH_3_), 2.25–2.18 (m, 1H, H-22b), 2.07–2.00 (m, 1H, H-6a), 2.00–1.95 (m, 1H, H-20), 1.84–1.77 (m, 3H, H-17, H-16), 1.72–1.66 (m, 2H, H-5, H-6b), 1.50–1.42 (m, 1H, H-1a), 1.27 (s, 3H, CH_3_), 1.25 (s, 3H, CH_3_), 1.11–1.09 (m, 9H, 3 ×CH_3_), 0.99 (s, 3H, CH_3_), 0.87 (d, *J* = 6.5 Hz, 2H). ^13^C-APT (150 MHz, CDCl_3_) *δ*: 217.2, 209.5, 199.6, 175.6, 159.3, 140.2, 136.1, 135.6, 129.3, 128.7, 72.4, 68.8, 53.9, 51.7, 49.9, 48.7, 48.1, 47.0, 46.8, 46.6, 40.8, 38.0, 36.4, 36.0, 35.5, 35.2, 34.3, 32.8, 29.0, 27.4, 21.1, 20.7, 19.6, 19.5, 19.4, 18.0, 17.3. HRMS calculated for C_39_H_55_NO_6_Na [M + Na]^+^ *m/z* 656.3922, found 656.3910.

#### 3.1.11. Synthesis of (4-Fluorophenethyl)-(7β,15α,25R)-7,15-Dihydroxy-3,11,23-Trioxolanost-8-en-26-oic Amide (**A11**)

The title compound was obtained from 4-fluorophenethylamine hydrochloride following similar synthesis procedure of **A1** (white powder, yield 97.2%). mp: 188.1–189.0 °C. ^1^H-NMR (600 MHz, CDCl_3_) *δ*: 7.09 (m, 2H, Ph-H), 6.92 (m, 2H, Ph-H), 5.83 (m, 1H, CO*N*H), 4.50 (m, 1H, H-15), 4.54 (m, 1H, H-7), 3.42–3.34 (m, 2H, CO*N*HCH_2_), 2.81–2.73 (m, 2H, H-1b, H-24a), 2.72–2.56 (m, 4H, H-12a, H-25, CH_2_Ph), 2.45–2.37 (m, 3H, H-22a, H-12b, H-24b), 2.36–2.28 (m, 2H, H-2), 2.19–2.10(m, 1H, H-22b), 1.99–1.94 (m, 1H, H-6a), 1.93–1.87 (m, 1H, H-20), 1.77–1.70 (m, 3H, H-17, H-16), 1.67–1.59 (m, 2H, H-5, H-6b), 1.44–1.35(m, 1H, H-1a), 1.20 (s, 3H, CH_3_), 1.18 (s, 3H, CH_3_), 1.04–1.02 (m, 9H, 3 × CH_3_), 0.91 (s, 3H, CH_3_), 0.79 (d, *J* = 6.4 Hz, 2H). ^13^C-APT (150 MHz, CDCl_3_) *δ*: 216.4, 208.6, 198.6, 174.7, 161.4, 159.8, 158.4, 139.0, 133.39, 133.37, 129.2, 129.2, 114.4, 114.3, 71.2, 67.7, 52.9, 50.6, 48.8, 47.6, 47.0, 46.0, 45.7, 39.7, 36.9, 35.2, 34.9, 34.4, 33.8, 33.2, 31.7, 27.9, 26.3, 19.6, 18.6, 18.5, 18.4, 17.0, 16.2. HRMS calculated for C_38_H_52_FNO_6_Na [M + Na]^+^ *m/z* 660.3671, found 660.3650.

#### 3.1.12. Synthesis of (4-Chlorophenethyl)-(7β,15α,25R)-7,15-Dihydroxy-3,11,23-Trioxolanost-8-en-26-oic Amide (**A12**)

The title compound was obtained from 4-chlorobenzeneethanamine following similar synthesis procedure of **A1** (white powder, yield 95.5%). mp: 195.4–196.1 °C. ^1^H-NMR (600 MHz, CDCl_3_) *δ*: 7.20 (m, 2H, Ph-H), 7.07 (m, 2H, Ph-H), 5.78 (m, 1H, CO*N*H), 4.70 (m, 1H, H-15), 4.55 (m, 1H, H-7), 3.99 (s, 1H, OH), 3.67 (m, 3H, OH, CO*N*HCH_2_), 2.81–2.73 (m, 2H, H-1b, H-24a), 2.72–2.55 (m, 4H, H-12a, H-25, CH_2_Ph), 2.45–2.37 (m, 3H, H-22a, H-12b, H-24b), 2.35–2.28 (m, 2H, H-2), 2.18–2.09(m, 1H, H-22b), 2.00–1.94 (m, 1H, H-6a), 1.93–1.87 (m, 1H, H-20), 1.78–1.70 (m, 3H, H-17, H-16), 1.66–1.59 (m, 2H, H-5, H-6b), 1.45–1.30 (m, 1H, H-1a), 1.19 (s, 3H, CH_3_), 1.18 (s, 3H, CH_3_), 1.04–1.02 (m, 9H, 3 × CH_3_), 0.91 (s, 3H, CH_3_), 0.79 (d, *J* = 6.5 Hz, 2H). ^13^C-APT (150 MHz, CDCl_3_) *δ*: 216.4, 208.6, 198.6, 174.7, 158.3, 139.1, 136.2, 131.3, 129.1, 127.7, 71.3, 67.7, 52.9, 50.6, 48.8, 47.6, 47.0, 46.0, 45.7, 45.6, 39.5, 36.9, 35.2, 34.9, 34.5, 34.0, 33.3, 31.7, 27.9, 26.3, 19.6, 18.6, 18.5, 18.4, 17.0, 16.2. HRMS calculated for C_38_H_53_ClNO_6_ [M + H]^+^ *m/z* 654.3556, found 654.3547.

#### 3.1.13. Synthesis of (4-Methylpiperazin-1-yl)-(7β,15α,25R)-7,15-Dihydroxy-3,11,23-Trioxolanost-8-en-26-oic Amide (**A13**)

The title compound was obtained from methylpiperazine following similar synthesis procedure of **A1** (white powder, yield 92.7%). mp: 173.2–174.9 °C. ^1^H-NMR (600 MHz, CDCl_3_) *δ*: 4.75–4.72 (m, 1H, H-15), 4.58–4.56 (m, 1H, H-7), 3.74–3.68 (m, 2H, CONCH_2_), 3.61−3.54 (m, 2H, CONCH_2_), 3.22–3.12 (1H, H-25), 3.01–2.95 (m, 1H, H-1b), 2.83–2.77 (m, 1H, H-24a), 2.72 (d, *J* = 16.5 Hz, 1H, H-12a), 2.64–2.51 (m, 2H, CH_2_N), 2.51–2.33 (m, 10H, H-24b, H-12b, H-22a, H-2, NCH_3_, CH_2_N), 2.25–2.16 (m, 1H, H-22b), 2.02–1.97 (m, 1H, H-6a), 1.97–1.91 (m, 1H, H-20), 1.82–1.73 (m, 3H, H-17, H-16), 1.71–1.62 (m, 2H, H-5, H-6b), 1.48–1.39 (m, 1H, H-1a), 1.24 (s, 3H, CH_3_), 1.22 (s, 3H, CH_3_), 1.09–1.06 (m, 9H, 3 × CH_3_), 0.98 (s, 3H, CH_3_), 0.84 (d, *J* = 6.5 Hz, 3H, CH_3_). ^13^C-APT (150 MHz, CDCl_3_) *δ*: 216.4, 208.8, 198.7, 173.1, 158.7, 138.9, 71.1, 67.6, 53.9, 53.5, 52.9, 50.7, 48.7, 47.6, 47.1, 46.2, 45.7, 45.6, 44.6, 44.1, 40.3, 36.9, 35.1, 34.5, 33.3, 31.8, 29.8, 27.8, 26.3, 19.7, 18.6, 18.5, 18.4, 16.5, 16.3. HRMS calculated for C_35_H_55_N_2_O_6_ [M + H]^+^
*m/z* 599.4055, found 599.4036.

#### 3.1.14. Synthesis of (4-Ethylpiperazin-1-yl)-(7β,15α,25R)-7,15-Dihydroxy-3,11,23-Trioxolanost-8-en-26-oic Amide (**A14**)

The title compound was obtained from 1-ethylpiperazine following similar synthesis procedure of A1 (white powder, yield 98.1%). mp: 198.5–199.4 °C. ^1^H-NMR (600 MHz, CDCl_3_) *δ*: 4.72−4.70 (m, 1H, H-15), 4.56−4.54 (m, 1H, H-7), 3.82–3.72 (m, 2H, CONCH_2_), 3.66−3.50 (m, 2H, CONCH_2_), 3.18–3.09 (1H, H-25), 2.98–2.90 (m, 1H, H-1b), 2.81–2.72 (m, 1H, H-24a), 2.72–2.66 (m, 3H, H-12a, CH_2_), 2.66–2.31 (m, 9H, H-24b, H-12b, H-22a, H-2, CH_2_N), 2.22–2.12 (m, 1H, H-22b), 1.99–1.94 (m, 1H, H-6a), 1.94–1.86 (m, 1H, H-20), 1.77–1.70 (m, 3H, H-17, H-16), 1.49–1.60 (m, 2H, H-5, H-6b), 1.45–1.35 (m, 1H, H-1a), 1.20 (s, 3H, CH_3_), 1.18 (s, 3H, CH_3_), 1.09 (t, *J* = 7.0 Hz, 3H, NCH_2_CH_3_), 1.05–1.02 (m, 9H, 3 × CH_3_), 0.92 (s, 3H, CH_3_), 0.80 (d, *J* = 6.2 Hz, 3H, CH_3_). ^13^C-APT (150 MHz, CDCl_3_) *δ*: 216.4, 208.9, 198.6, 173.2, 158.4, 139.0, 71.2, 67.7, 52.9, 51.5, 51.2, 51.0, 50.7, 48.8, 47.6, 47.1, 46.2, 45.7, 45.6, 43.7, 40.0, 36.9, 35.2, 34.5, 33.3, 31.8, 29.7, 27.9, 26.4, 19.7, 18.6, 18.5, 18.4, 16.5, 16.3. HRMS calculated for C_36_H_57_N_2_O_6_ [M + H]^+^ *m/z* 613.4211, found 613.4205.

#### 3.1.15. Synthesis of (4-Phenylpiperazin-1-yl)-(7β,15α,25R)-7,15-Dihydroxy-3,11,23-Trioxolanost-8-en-26-oic Amide (**A15**)

The title compound was obtained from 1-phenylpiperazine following similar synthesis procedure of **A1** (white powder, yield 89.9%). mp: 229.4–229.9 °C. ^1^H-NMR (600 MHz, CDCl_3_) *δ*: 7.23−7.21 (m, 2H, Ph-H), 6.88−6.83 (m, 3H, Ph-H), 4.72−4.69 (m, 1H, H-15), 4.55−4.53 (m, 1H, H-7), 3.75−3.71 (m, 2H, CONCH_2_), 3.65−3.62 (m, 2H, CONCH_2_), 3.26–3.16 (m, 2H, CH_2_), 3.17–3.10(1H, H-25), 3.10–3.05 (m, 2H, CH_2_), 3.00–2.92 (m, 1H, H-1b), 2.80–2.74 (m, 1H, H-24a), 2.78 (d, *J* = 16.5 Hz, 1H, H-12a), 2.45–2.39 (m, 3H, H-22a, H-12b, H-24b), 2.38–2.31 (m, 2H, H-2), 2.22–2.14 (m, 1H, H-22b), 2.00–1.94 (m, 1H, H-6a), 1.94–1.88 (m, 1H, H-20), 1.78–1.71 (m, 3H, H-17, H-16), 1.66–1.58 (m, 2H, H-5, H-6b), 1.43–1.35 (m, 1H, H-1a), 1.20 (s, 3H, CH_3_), 1.18 (s, 3H, CH_3_), 1.06 (d, *J* = 7.1 Hz, 3H, CH_3_), 1.04 (s, 3H, CH_3_), 1.02 (s, 3H, CH_3_), 0.92 (s, 3H, CH_3_), 0.80 (d, *J* = 6.4 Hz, 3H, CH_3_). ^13^C-APT (150 MHz, CDCl_3_) *δ*: 216.3, 208.6, 198.6, 173.1, 158.2, 149.8, 139.1, 128.2, 119.5, 115.5, 71.3, 67.7, 52.9, 50.6, 48.8, 48.7, 48.4, 47.6, 47.1, 46.2, 45.7, 45.6, 44.6, 40.9, 37.6, 36.9, 35.3, 34.4, 33.2, 31.7, 29.9, 27.9, 26.3, 19.6, 18.6, 18.5, 18.4, 16.6, 16.2. HRMS calculated for C_40_H_57_N_2_O [M + H]^+^ *m/z* 661.4211, found 661.4203.

### 3.2. Cell Culture

The cell lines present in this study were obtained from Procell Life Science & Technology Co. Ltd. MCF-7, HepG2 and SJSA-1 cells were cultured in DMEM medium (DMEM, Gibco) supplemented with 10% fetal bovine serum (FBS, Gibco), 1% penicillin-streptomycin (Hyclone) at 37℃ in a humid environment with 5% CO_2_. HK-2 cells were cultured in DMEM/F12 (1:1) medium and placed in incubators in the same environment.

### 3.3. Cell Viability Assay

Cell viability was determined by MTT assay. MCF-7, HepG2, SJSA-1 and HK-2 cells (6 × 10^3^ cells /well) were seeded in 96-well plates with serum-free medium for 24 h. Then MCF-7, HepG2, SJSA-1 cells were treated with 0.1% DMSO, 25, 50, 100 μM of GAA derivatives for 48 h (MCF-7, HepG2, HK2) or 72 h (SJSA-1). After 48 or 72 h, 10 μL MTT (5 mg/mL, Beyotime) was added and incubated at 37 °C for 4 h. Then 100 μL of lysate was added. After complete dissolution of the crystal, the absorbance was measured at 540 nm and expressed as the average percentage of absorbance between treated and control cells. The value for control cells was set at 100%. Cell survival rate was calculated as the ratio of the absorbance of the cells and negative control after minus the blank absorbance respectively.

### 3.4. Target Fishing and Molecular Docking by In Silico Approaches

The binding targets of GAA were predicted using Discovery Studio 2016 v16.1 (BIOVIA Software Inc., San Diego, CA, USA), a software suite for the computational analysis of data relevant to Life Sciences research. To predict the probable targets of GAA, we used Ligand Profiler protocol which maps a set of pharmacophores, including Pharma DB by default. The ligand GAA was prepared by the Specifying Ligands parameter protocol. After setting parameters, the job was run, and the results were gained for three days. To explore the potential binding mode of GAA and **A2** with MDM2 protein (PDB code: 4j3e), a molecular modeling research was performed with docking program named Induced-Fit, a refinement method in another software MOE. To eliminate any bond length and bond angle biases, the ligands (GAA and **A2**) were subjected to the “energy minimize” prior to docking. The binding affinities (*S*-values) in MOE were used to evaluate the interactions between MDM2 and ligands. The scores (binding affinities) were obtained based on the virtual calculation of various interactions of ligands with the targeted receptor.

### 3.5. Surface Plasmon Resonance (SPR) Assay

GAA derivatives bound to MDM2 protein were assayed with a molecular interaction analyzer (PALL FORTEBIO, USA). MDM2 protein (5 mg/mL, Protintech) was immobilized on a PCH sensor chip (Octet) and preactivated with EDC/NHS mixture for 420 s at a flow rate of 10 μL/min. **A2** was diluted to 100, 50, 25, 12.5, 6.25, 3.13, 0 μM with PBST buffer containing 1% DMSO. The binding time was 600 s, and the flow rate was 20 μL/min. The dissociation time was 180 s, and the affinity constant K_D_ value was obtained by computer fitting and steady-state analysis.

### 3.6. Flow Cytometric Analysis of the Apoptosis Rate with Annexin V-FITC/PI Staining

To determine the apoptosis rate, an Annexin V-FITC/PI double staining apoptosis assay kit (Beyotime) was used to detect apoptotic cells by flow cytometry (BD FACSALOBUR), according to the manufacturer’s instructions. Briefly, SJSA-1 cells were treated with 0.1% DMSO, 12.5, 25, and 50 μM of **A2** for 24 h. After harvesting, the cells were incubated with 5 μL Annexin V-FITC for 15 min and 10 μL PI for 5 min at 4 °C under dark conditions. Flow cytometry was then performed to analyze the apoptosis rate. Data were analyzed by using BD FACSDiva 8.0.1.

### 3.7. Western Blot Analysis of Protein Expression

For Western blot analysis, MCF-7 and SJSA-1 cells were treated with different concentrations of **A2** for 24 h. The total cell protein was extracted, and proteins were isolated using 10% SDS-PAGE gel system. The proteins on the gel were transferred to PVDF membrane, blocked in 5% BSA at room temperature for 2 h, incubated in primary antibody dilution at 4 °C overnight, and washed with TBST for 3 times, 10 min each. Then, they were transferred to dilute release solution of secondary antibody and incubated at room temperature for 2 h. ECL chemiluminescence development solution (Beyotime, BeyoECL star) was added uniformly and detected on gel imaging system (Clinx ChemiScope, China). Antibodies for blotting were MDM2 (abcam, ab16895), P53 (Proteintech, 10442-1-AP), Bcl-2 (CST, 15071S), Bax (CST, 2772T) and β-actin (abcam, ab8226).

### 3.8. Statistical Analysis

The results are expressed as the means ± standard deviation. A one-way AVONA and t-test were used for comparison of differences between groups, and GraphPad Prism 8.0 software was used for graph and statistical analysis. Statistical significance was set at *p* < 0.05.

## 4. Conclusions

Natural products are rich in beneficial scaffolds that have been used in anti-tumor, anti-inflammatory, neuroprotective and other aspects. However, these natural products have the problem of unclear targets and weak activity. Therefore, if we can determine the relevant mechanism of the action of natural products and identify specific pathways and targets, we can improve their activity based on adding appropriate interaction with binding amino acid residues in the active pocket of the target. In this study, we modified *Ganoderma lucidum* triterpenoid compound GAA and evaluated anti-proliferative effects of these derivatives in different tumor cell lines. Finally, compound **A2** was selected for further investigation of its mechanism. The results showed that **A2** could induce apoptosis by interfering with the MDM2-p53 pathway. Target fishing and SPR experiments suggested that **A2** might play a role by binding to MDM2 and blocking its inhibition of p53. Although these compounds may have weaker anti-tumor activity than other small molecule anti-tumor drugs, this study may provide insights into finding the target of GAA and developing new natural product anti-cancer compounds. If we can confirm the specific targets of GAA in different diseases, we can carry out target-based rational design of GAA to greatly improve its efficacy and provide an excellent scaffold for the development of new drugs.

## 5. Patents

In order to protect the structure and activity of compounds in a timely manner, the patent Preparation method of Ganoderic A amide derivatives useful as anti-tumor drugs, China CN112574272 A 2021-03-30, refers to the synthesis of the derivatives and simple in vitro cell anti-proliferation screening. In subsequent studies, the activity of the derivatives in other cell lines was found and the mechanism was investigated. The relevant experimental results are presented in this article.

## Data Availability

Not applicable.

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
