# Peer review of "Ganoderic Acid A and Its Amide Derivatives as Potential Anti-Cancer Agents by Regulating the p53-MDM2 Pathway: Synthesis and Biological Evaluation"

_molecules, 2023, doi:10.3390/molecules28052374_

Round 1

Reviewer 1 Report (Previous Reviewer 1)

The revised version has been significantly improved. I consider this work attracts sufficient readership for Molecules and thus recommend it for publication.

Author Response

Dear Reviewer,

We sincerely thank you for your appreciated advice and giving us extremely valuable suggestions. 

With best regards,

Yu Tian, Ph.D.

Professor of Medicinal chemistry

Institute of Medicinal Plant Development

Chinese Academy of Medical Sciences & Peking Union Medical College

Beijing 100193, P.R. China

Reviewer 2 Report (Previous Reviewer 3)

The publication written by Jia, Li, Shang, Luo, and Tian has been greatly improved and corrected from the previously reviewed original version, and in my opinion may be oppressive in Molecules. The authors followed most of the previous comments.

Author Response

Dear Reviewer,

We sincerely thank you for your appreciated advice and giving us extremely valuable suggestions. 

With best regards,

Yu Tian, Ph.D.

Professor of Medicinal chemistry

Institute of Medicinal Plant Development

Chinese Academy of Medical Sciences & Peking Union Medical College

Beijing 100193, P.R. China

Reviewer 3 Report (New Reviewer)

The article “Ganoderic Acid A and its amide Derivatives as Potential Anti-cancer Agents by Regulating p53-MDM2 Pathway: Synthesis and Biological Evaluation” reported by Yi Jia and the group explores the synthesis and antitumor activity of GAA against various cancer cell lines and also provided a mode of action. Although the compounds were not very potent but the authors have provided enough biological studies which help the researcher for the future design of the molecules in this direction. This manuscript can be accepted after addressing the minor comments below.

1.      Structures of the molecules in fig. 1 and 2 were very small.

2.      Reaction conditions in scheme 1 were incomplete e. g temp. and reaction time was missing.

3.      Authors need to add some references in the synthetic part.

4.      Please provide the key NMR and HRMS features of one of the synthesized compounds in the result and discussion section of the chemistry part.

5.      Molecular docking needs more explanation in terms of binding energies or docking scores.

6.      What solvent system was used to purify the compounds?

7.      Please provide the Melting points of all the newly synthesized compounds.

8.      The NMR data provided in the manuscript did not match the supplementary information.

9.      Why the peaks in the 13C NMR were downward? Is it 13C NMR or Dept? The authors should clarify this confusion.

10.  References no. 3 and 18 were not according to journal guidelines.

Author Response

Dear Managing Editor and Reviewers,

On behalf of my co-authors, we thank you very much for giving us an opportunity to revise our manuscript, we appreciate editor and reviewers very much for their positive and constructive comments and suggestions on our manuscript entitled “Ganoderic Acid A and Its Amide Derivatives as Potential Anti-cancer Agents by Regulating p53-MDM2 Pathway: Synthesis and Biological Evaluation”. (ID: 2238873).

We have studied reviewer’s comments carefully and have made revision which marked in revisions mode. We have tried our best to revise our manuscript according to the comments. Here is our response to reviewer 3.

Q1: Structures of the molecules in fig. 1 and 2 were very small.

A1: Thank you for your comments. We have changed the size of the structure in Figures 1 and 2.

Q2: Reaction conditions in scheme 1 were incomplete e. g temp. and reaction time was missing.

A2: Thank you for pointing out the question. We added the solvent, temperature and reaction time in scheme 1. The reaction conditions are also proposed in the caption.

Q3: Authors need to add some references in the synthetic part.

A3: Thank you for your suggestion. We add the references [28, 29] using the same reaction conditions in line 116. The references are showed below.

[28] Walczak, J.; Maksymilian, D.; Ziomkowska, M.; Śliwka-Kaszyńska, M.; Daśko, M.; Trzonkowski, P.; Cholewiński, Grzegorz. Novel amides of mycophenolic acid and some heterocyclic derivatives as immunosuppressive agents. J. Enzyme. Inhib. Med. Chem. 2022, 37(1), 2725-2741. doi:10.1080/14756366.2022.2127701.

[29] Nassim. E.; Wang, M. Synthesis and biological evaluation of naloxone and naltrexone-derived hybrid opioids. Med. Chem. 2012, 8(4), 683-689. doi:10.2174/157340612801216193.

Q4: Please provide the key NMR and HRMS features of one of the synthesized compounds in the result and discussion section of the chemistry part.

A4: Thank you for your comment. We added the key NMR features in line 123 as “After substituted by different amine fragments, the hydrogen signal of amide bond appears at 7-5 ppm. The methylene peak of amine fragments is mostly distributed at 4.5-3 ppm. The chemical shift of hydrogen signal in GAA itself does not change very much.”

Q5: Molecular docking needs more explanation in terms of binding energies or docking scores.

A5: Thank you for your comment. We added the relevant explanation in line 231 as “We use the S-value to evaluate the binding degree of the compound and MDM2 in the molecular docking experiments. The higher the absolute value of this number, the stronger the binding force.”.

Q6: What solvent system was used to purify the compounds?

A6: The solvent system is dichloromethane and methanol (10:1, V/V). Thank you for pointing out this question. We add the information in line 286 as “The crude material was purified by column chromatography using dichloromethane and methanol (10:1, V/V) as mobile phase to obtain target molecule as white powder (yield 80.6%)”.

Q7: Please provide the Melting points of all the newly synthesized compounds.

A7: Thank you for your comment. We measured the melting points of the compound and present the results in 4.1 Chemistry part.

Q8: The NMR data provided in the manuscript did not match the supplementary information.

A9: Thank you for your question. After carefully checking the corresponding sequence of the compound and the spectrum and finding no errors, we speculate that the mismatch you are referring to is that we have written the 13C-APT spectrum into 13C NMR. We are very sorry to make these small mistakes, and have changed the relevant statement.

Q9: Why the peaks in the 13C NMR were downward? Is it 13C NMR or Dept? The authors should clarify this confusion.

A9: Sorry for causing confusion. The spectrum is 13C-APT. We changed the relevant statement in the manuscript.

Q10: References no. 3 and 18 were not according to journal guidelines.

A10: We are very sorry for these minor errors. We have revised the format of references no. 3 and 18 and carefully checked other references to ensure that there are no other errors.

We would like to express our great appreciation to you and reviewers for comments on our paper. Looking forward to hearing from you.

Thank you and best regards.

Sincerely yours,

Yu Tian, Ph.D.

Professor of Medicinal chemistry

Institute of Medicinal Plant Development

Chinese Academy of Medical Sciences & Peking Union Medical College

Beijing 100193, P.R. China

Tel: +86-10-57833018;

FAX: +86-10-57833018;

E-mail: [email protected]; [email protected]

This manuscript is a resubmission of an earlier submission. The following is a list of the peer review reports and author responses from that submission.

Round 1

Reviewer 1 Report

In this manuscript, Tian and co-workers report the synthesis and anticancer activity evaluation of the amide derivatives of ganoderic acid A, and further studies on the mechanism of action of  compound A2. It is a topic of interest to the researchers in the related areas but the paper needs very significant improvement before acceptance for publication.

My detailed comments are as follows:

1.  This manuscript needs careful editing by someone with expertise in technical English editing  so that the goals and results of the study are clear to the reader. Some sentences contain grammatical mistakes or are not complete sentences, such as,

  Line_31: "Ganoderma acid A" would be "Ganoderic acid A"

  Line_46, 49:  "p53" would be "P53". 

  Line_70 "proteasome" would be  "proteosome"

  In Table 1. All compounds' name should be in bold.

  Line_279 "H-NMR" would be "1H-NMR"

2. Chiral configrations of C20, C25 should be shown in Scheme 1 and Table 1.

3. In page_2, line_46~58, the description of MDM2-p53 signaling pathway is  a little confusing. Is p53 a protein or a gene?

4. What is the relationship of MDM2 and p53? If the MDM2 is a inhibitor of p53, then the result of "both MDM2 and P53 showed an increasing trend" is not reasonable.

5. The molecular formula of HRMS should be consistent with the results.

    For example, in line_273, the Na should be included in the molecular formula.

6. It is difficult to conclude from Figure 6 that “after treating MCF-7 cells with different concentrations of A2 for 24 hours, both MDM2 and P53 showed an increasing trend”.  The relative protein expression analysis should be added.

7. The list of references is not in Molecules style. It is close but not completely correct.

8.  Please submit a new 1H NMR spectra with a range of 0-10 ppm for all compounds. 

Author Response

Response to Reviewer 1 Comments

The questions raised by the Referee 1 are answered as follows:

Comments and Suggestions for Authors

In this manuscript, Tian and co-workers report the synthesis and anticancer activity evaluation of the amide derivatives of ganoderic acid A, and further studies on the mechanism of action of compound A2. It is a topic of interest to the researchers in the related areas but the paper needs very significant improvement before acceptance for publication.

My detailed comments are as follows:

  1. This manuscript needs careful editing by someone with expertise in technical English editing so that the goals and results of the study are clear to the reader. Some sentences contain grammatical mistakes or are not complete sentences, such as,

  Line_31: "Ganoderma acid A" would be "Ganoderic acid A"

  Line_46, 49: "p53" would be "P53".

  Line_70 "proteasome" would be "proteosome"

  In Table 1. All compounds' name should be in bold.

  Line_279 "H-NMR" would be "1H-NMR"

  1. Chiral configrations of C20, C25 should be shown in Scheme 1 and Table 1.
  2. In page_2, line_46~58, the description of MDM2-p53 signaling pathway is a little confusing. Is p53 a protein or a gene?
  3. What is the relationship of MDM2 and p53? If the MDM2 is a inhibitor of p53, then the result of "both MDM2 and P53 showed an increasing trend" is not reasonable.
  4. The molecular formula of HRMS should be consistent with the results.

  For example, in line_273, the Na should be included in the molecular formula.

  1. It is difficult to conclude from Figure 6 that “after treating MCF-7 cells with different concentrations of A2 for 24 hours, both MDM2 and P53 showed an increasing trend”. The relative protein expression analysis should be added.
  2. The list of references is not in Molecules style. It is close but not completely correct.
  3. Please submit a new 1H NMR spectra with a range of 0-10 ppm for all compounds.

Responses:

Thank you very much for taking the time to review this manuscript and put forward your valuable comments. The following is the answer and explanation of your comments.

Q1. This manuscript needs careful editing by someone with expertise in technical English editing so that the goals and results of the study are clear to the reader. Some sentences contain grammatical mistakes or are not complete sentences, such as,

  Line_31: "Ganoderma acid A" would be "Ganoderic acid A"

  Line_46, 49: "p53" would be "P53".

  Line_70 "proteasome" would be "proteosome"

  In Table 1. All compounds' name should be in bold.

  Line_279 "H-NMR" would be "1H-NMR"

A1. Thank you for your kind suggestion. In accordance with your proposed changes, "Ganoderma acid A" was changed to "Ganoderic acid A". The “p53” in this article was carefully checked and relevant sentences were revised to avoid ambiguity. Other spelling errors such as "proteasome" and "H-NMR" were also revised. Thank you for pointing out these English writing issues. Our manuscript has been reviewed by a native English speaker, please see the revised manuscript for details of the changes. Changes may be reflected in the revision track.

Q2. Chiral configrations of C20, C25 should be shown in Scheme 1 and Table 1.

A2. Thank you for pointing out the carelessness of the chiral configurations in Table 1 and Scheme 1. We have revised them.

Scheme 1. The synthesis of GAA derivatives. Reagents and conditions: TBTU, DIPEA, DCM, rt, 0.5-2 h.

Q3. In page_2, line_46~58, the description of MDM2-p53 signaling pathway is  a little confusing. Is p53 a protein or a gene?

A3. p53 is a gene and P53 is the protein that p53 expressed. We have changed the relevant description in this segment and added the MDM2-p53 signaling pathway diagram to explain the relationship between MDM2 and p53.

The added part is “If too much p53 protein is produced during certain physiological processes, cell function is impaired or the tendency to form tumors is increased. Therefore, the expression of Murine Double Minute 2 (MDM2) protein in the downstream signaling pathway will rise when P53 accumulates in normal cells. To achieve the balance and stability of P53 levels in cells, MDM2 can interact with the transcriptional activation domain of the p53 gene to form the p53-MDM2 complex, which suppresses the transcriptional activity of p53. When a cell is stressed, MDM2 expression decreases, p53 expression increases, and the increase in p53 induces MDM2 expression at the transcriptional level, creating a negative feedback regulatory loop (Figure 2) [19]. MDM2 can also act as an E3 ubiquitin ligase, targeting P53 and inducing its ubiquitination and degradation to maintain low levels of P53 [20]”.

Figure 2. MDM2-p53 negative feedback regulation loop.

The relevant literature [19, 20] is also added.

19.Zhao, Y.; Yu, H.; Hu, W. The regulation of MDM2 oncogene and its impact on human cancers. Acta Biochim. Biophys. Sin 2014, 46, 180-189. doi: 10.1093/abbs/gmt147.

20.Li, B.; Cheng, Q.; Li, Z.; Chen, J. p53 inactivation by MDM2 and MDMX negative feedback loops in testicular germ cell tumors. Cell cycle (Georgetown, Tex.) 2010, 9, 1411-1420. doi: 10.4161/cc.9.7.11255.

Q4. What is the relationship of MDM2 and p53? If the MDM2 is a inhibitor of p53, then the result of "both MDM2 and P53 showed an increasing trend" is not reasonable.

A4. The MDM2 is an inhibitor of p53. Since MDM2-p53 forms a negative feedback regulation, the inhibition of MDM2 can induce the activation of p53. Activation of p53 results in transcription of MDM2 mRNA, leading to robust accumulation of MDM2 protein. So there is an increase in both MDM2 and P53, which is also a drawback of MDM2 inhibitors. Once the MDM2 inhibitor is cleared due to the pharmacokinetic effect, the accumulated MDM2 in vivo can efficiently and rapidly degrade p53. These have been added to the Introduction mentioned above. We have also added the explanation in 2.4.2 as “And by blocking the interaction of MDM2 and p53, the activation of p53 results in transcription of MDM2 mRNA, leading to robust MDM2 protein accumulation”.

Q5. The molecular formula of HRMS should be consistent with the results.

A5. Thank you for pointing out the issue about molecular formula of HRMS. We have revised them.

Q6. It is difficult to conclude from Figure 6 that “after treating MCF-7 cells with different concentrations of A2 for 24 hours, both MDM2 and P53 showed an increasing trend”. The relative protein expression analysis should be added.

A6. Due to the pandemic and time constraints, we were unable to repeat the WB experiment more than three times, so it may be not be rigorous to include the histogram results of the gray intensity analysis in this article. Therefore, we changed our statement in the article from “after treating MCF-7 cells with different concentrations of A2 for 24 hours, both MDM2 and P53 showed an increasing trend” to “after treating MCF-7 cells with A2 for 24 hours, both MDM2 and P53 showed an increasing trend at 50 µM”.

Q7. The list of references is not in Molecules style. It is close but not completely correct.

A7. Sorry for the carelessness of the reference style. We have revised them.

Q8. Please submit a new 1H NMR spectra with a range of 0-10 ppm for all compounds.

A8. We have updated the 1H NMR spectra with a range of 0-10 ppm for all compounds in the supplementary material. There is an example for A1.

Reviewer 2 Report

This work by Jia and coworkers describes the synthesis of ganoderic acid A derivatives and their antiproliferative activity in selected cancer cell lines. Overall the manuscript could have some interest for the readers of this journal, but I think that in the current form it is not ready for publication in Molecules.

The main flaws are the following:

1) The background of this study is not solid. Authors report that "GAA may interact with P53-MDM2 pathway to some extent", and "GAA is probably related to P53-MDM2 pathway". In this situation, in my opinion it is not sufficient speculate in silico the activity of these compounds (by target phishing and docking experiments), but computational methods have to be validated by in vitro experiments.

2) It is unclear the rational design of these derivatives of GAA. Why the authors selected the amides? Is this choice casual or driven by chemical reasons?

3) The antiproliferative activity of compounds is expressed as percentage of cell viability. I think that the authors at least should calculate IC50s. The observed antiproliferative effects are at high concentrations, and they do not sound so interesting, also if values are improved with respect to GAA. I am not convinced about the real interest in this class of compounds, whose anticancer activity is only detected at high concentrations.

For these reasons, I repute that this article should be rejected in the current form.

Author Response

Response to Reviewer 2 Comments

The questions raised by the Referee 2 are answered as follows:

Comments and Suggestions for Authors

This work by Jia and coworkers describes the synthesis of ganoderic acid A derivatives and their antiproliferative activity in selected cancer cell lines. Overall the manuscript could have some interest for the readers of this journal, but I think that in the current form it is not ready for publication in Molecules.

The main flaws are the following:

1) The background of this study is not solid. Authors report that "GAA may interact with P53-MDM2 pathway to some extent", and "GAA is probably related to P53-MDM2 pathway". In this situation, in my opinion it is not sufficient speculate in silico the activity of these compounds (by target phishing and docking experiments), but computational methods have to be validated by in vitro experiments.

2) It is unclear the rational design of these derivatives of GAA. Why the authors selected the amides? Is this choice casual or driven by chemical reasons?

  • The antiproliferative activity of compounds is expressed as percentage of cell viability. I think that the authors at least should calculate IC50s. The observed antiproliferative effects are at high concentrations, and they do not sound so interesting, also if values are improved with respect to GAA. I am not convinced about the real interest in this class of compounds, whose anticancer activity is only detected at high concentrations.

Responses:

Thank you very much for taking the time to review this manuscript and put forward your valuable comments. The following is the answer and explanation of your comments.

Q1: The background of this study is not solid. Authors report that "GAA may interact with P53-MDM2 pathway to some extent", and "GAA is probably related to P53-MDM2 pathway". In this situation, in my opinion it is not sufficient speculate in silico the activity of these compounds (by target phishing and docking experiments), but computational methods have to be validated by in vitro experiments.

A1: Thank you for your question. After analysing the in silico experimental results, we found that GAA and MDM2 have certain interaction, and the possibility of the interaction between GAA and MDM2 can be proved by relevant literature. Then we used SPR experiment to find that GAA and MDM2 do have certain affinity in vitro, and WB results also meet our expectation, so we speculate that GAA can activate p53 by inhibiting MDM2. However, due to the limitations of experimental conditions, we did not perform the co-crystal structure study of the complex and other in vitro experiments to verify the interaction between MDM2 and GAA. We will further verify the interaction between GAA and MDM2 in future research.

Q2: It is unclear the rational design of these derivatives of GAA. Why the authors selected the amides? Is this choice casual or driven by chemical reasons?

A2: The initial design of these molecules is based on preserving the core structure of GAA while enhancing its anti-tumor activity at the same time, so as to facilitate the subsequent study of its potential target. Therefore, the carboxyl is selected for modification. The selected amines consist of amine compounds with different length, substituent and polarity, such as fatty amine, aniline, benzylamine, phenylethylamine, and piperazine, in order to discuss the structure-activity relationship. Therefore, these amine compounds are selected.

Q3: The antiproliferative activity of compounds is expressed as percentage of cell viability. I think that the authors at least should calculate IC50s. The observed antiproliferative effects are at high concentrations, and they do not sound so interesting, also if values are improved with respect to GAA. I am not convinced about the real interest in this class of compounds, whose anticancer activity is only detected at high concentrations.

A3: It is a pity that, as you said, the anti-tumor activity of these compounds is not as high as that of small molecule anti-tumor drugs, but they are much better than the GAA. This study is more focused on discussing the effect of GAA and its derivatives on MDM2. Therefore, from these derivatives, we selected A2 with potent effects on different tumor cell lines to study the relevant mechanism. Due to the relatively weak activity of these compounds and the main aim of this article, we did not calculate IC50.

Reviewer 3 Report

The manuscript contributed by Yi Jia and co-authors titled “Synthesis and anticancer activity evaluation of ganoderic acid a derivatives as the potential MDM2 inhibitors” is interesting and scientifically valuable. However, in order for it to be published, the Authors should correct and clarify a few issues.

The HRMS spectra raise the greatest doubts. Are you sure it is HR spectra? Modern HR MS analyses aim to have only one peak in the spectrum, the M+H or M+Na molecular peak. The spectra contain a lot of peaks, which shows the low purity of the tested substances. The authors should ensure that the apparatus on which the analyses were conducted is the HR or HS apparatus.

Melting points are missing for a full characterisation of the described compounds. Such data would also allow conclusions about the purity of the substance.

In addition, Figure 1 and Scheme 1 should be larger because they are very illegible. Additionally, if substances derived from substance A are numbered A1-A12, why are derivatives of substance B numbered A12-A15 and not B1-B3. This introduces unnecessary doubts.

Author Response

Response to Reviewer 3 Comments

The questions raised by the Referee 3 are answered as follows:

Comments and Suggestions for Authors

The manuscript contributed by Yi Jia and co-authors titled “Synthesis and anticancer activity evaluation of ganoderic acid a derivatives as the potential MDM2 inhibitors” is interesting and scientifically valuable. However, in order for it to be published, the Authors should correct and clarify a few issues.

The HRMS spectra raise the greatest doubts. Are you sure it is HR spectra? Modern HR MS analyses aim to have only one peak in the spectrum, the M+H or M+Na molecular peak. The spectra contain a lot of peaks, which shows the low purity of the tested substances. The authors should ensure that the apparatus on which the analyses were conducted is the HR or HS apparatus.

Melting points are missing for a full characterisation of the described compounds. Such data would also allow conclusions about the purity of the substance.

In addition, Figure 1 and Scheme 1 should be larger because they are very illegible. Additionally, if substances derived from substance A are numbered A1-A12, why are derivatives of substance B numbered A12-A15 and not B1-B3. This introduces unnecessary doubts.

Responses:

Thank you very much for taking the time to review this manuscript and put forward your valuable comments. The following is the answer and explanation of your comments.

Q1: The HRMS spectra raise the greatest doubts. Are you sure it is HR spectra? Modern HR MS analyses aim to have only one peak in the spectrum, the M+H or M+Na molecular peak. The spectra contain a lot of peaks, which shows the low purity of the tested substances. The authors should ensure that the apparatus on which the analyses were conducted is the HR or HS apparatus.

A1: Thank you for your question. We have reconfirmed that it is HRMS. Other peaks could be ionic fragments due to high voltage. And in the Supplementary Material you can see that A1’s NMR spectra are pure. We can also find this situation in other articles. The above is an example from https://doi.org/10.1021/acs.jmedchem.1c02221 (Rational Design for Nitroreductase (NTR)-Responsive Proteolysis Targeting Chimeras (PROTACs) Selectively Targeting Tumor Tissues ).

Q2: Melting points are missing for a full characterisation of the described compounds. Such data would also allow conclusions about the purity of the substance.

A2: Thank you for your suggestion. However, due to the epidemic and the lack of experimental equipment, we have not been able to measure the melting point. The time for revising the article is close to the Chinese Lunar New Year, so school is out. The time for revision is too short to wait until the melting point is added after the school starts. Since this article will determine whether I can successfully graduate, I appreciate your understanding of this situation. Thank you for your valuable suggestions. In the following experiment we will measure the melting point of the compound.

Q3: In addition, Figure 1 and Scheme 1 should be larger because they are very illegible. Additionally, if substances derived from substance A are numbered A1-A12, why are derivatives of substance B numbered A12-A15 and not B1-B3. This introduces unnecessary doubts.

Q3: Thank you for your suggestion. We have changed the size of the relevant pictures. And the ambiguous parts in Scheme 1 have been revised.

Scheme 1. The synthesis of GAA derivatives. Reagents and conditions: TBTU, DIPEA, DCM, rt, 0.5-2 h.

Reviewer 4 Report

Manuscript ID molecules-2177314 entitled "Synthesis and Anticancer Activity Evaluation of Ganoderic acid A Derivatives as the Potential MDM2 Inhibitors". This study is well analyzed and well written manuscript describing the results of Anticancer Activity Evaluation of Ganoderic acid in which tests were provided and addresses a topic of major interest for anti-tumor drug development. I would recommend its publication in molecules.

Some suggestions here after.

1.      Ganoderma lucidum should be kept in italics throughout the text.

2.      In Abstract line 11: Its extensive pharmacological activity- this should be changed to the multitherapeutic potential of GAA,

3.      In Abstract the author should incorporate some results

4.      Figure 1. The structure of GAA and compound A2 should be revised to the chemical structure of GAA and GA derivative A2. Also, it is advised to update with high-quality images using Chemdraw or similar software.

5.      Please provide more information about the overall strategy for screening and treatment with statistical tests used.

6.      At the end of the abstract and conclusion by saying "It provides some ideas for the research on the target and anti-tumor mechanism of Ganoderic acid compounds and the development of related anti-tumor candidates." In view of this, how definitive are your results, and how applicable clinically?

7.      Limitations and future prospectus of the present work should be added after the conclusion section.

8.      I would encourage authors to provide the mechanistic approach as Graphical Abstract that would provide the thorough insights into the present investigation.

9.      In Supplementary files the quality of image is better than the MS file. Please provide the high-quality images in the MS. 

Author Response

Response to Reviewer 4 Comments

The questions raised by the Referee 4 are answered as follows:

Comments and Suggestions for Authors

Manuscript ID molecules-2177314 entitled "Synthesis and Anticancer Activity Evaluation of Ganoderic acid A Derivatives as the Potential MDM2 Inhibitors". This study is well analyzed and well written manuscript describing the results of Anticancer Activity Evaluation of Ganoderic acid in which tests were provided and addresses a topic of major interest for anti-tumor drug development. I would recommend its publication in molecules.

Some suggestions here after.

  1. Ganoderma lucidum should be kept in italics throughout the text.
  2. In Abstract line 11: Its extensive pharmacological activity- this should be changed to the multitherapeutic potential of GAA,
  3. In Abstract the author should incorporate some results
  4. Figure 1. The structure of GAA and compound A2 should be revised to the chemical structure of GAA and GA derivative A2. Also, it is advised to update with high-quality images using Chemdraw or similar software.
  5. Please provide more information about the overall strategy for screening and treatment with statistical tests used.
  6. At the end of the abstract and conclusion by saying "It provides some ideas for the research on the target and anti-tumor mechanism of Ganoderic acid compounds and the development of related anti-tumor candidates." In view of this, how definitive are your results, and how applicable clinically?
  7. Limitations and future prospectus of the present work should be added after the conclusion section.
  8. I would encourage authors to provide the mechanistic approach as Graphical Abstract that would provide the thorough insights into the present investigation.

  1. In Supplementary files the quality of image is better than the MS file. Please provide the high-quality images in the MS.

Responses:

Thank you very much for taking the time to review this manuscript and put forward your valuable comments. The following is the answer and explanation of your comments.

Q1: Ganoderma lucidum should be kept in italics throughout the text.

A1: Thank you for your suggestion. We have revised them.

Q2: In Abstract line 11: Its extensive pharmacological activity- this should be changed to the multitherapeutic potential of GAA

A2: Thank you for your suggestion. We have changed this statement to “The multitherapeutic potential of GAA, particularly its antitumor activity, has been widely studied”.

Q3: In Abstract the author should incorporate some results

A3: Thank you for your suggestion. As anti-proliferation activity is presented in terms of cell viability, we have included the computational results and SPR results in the abstract. The relative statement is changed to “In this study, we found that GAA has a certain interaction with the MDM2 protein in an in silico study (S-value: -6.49). GAA was then modified to synthesize a series of amide compounds on carboxyl group, and their in vitro anti-tumor activities were investigated. Finally, compound A2 was selected to study its mechanism of action because of its high activity in three different types of tumour cell lines and low toxicity to normal cells. The results showed that A2 could induce apoptosis via the p53 signaling pathway, and it may be involved by binding to MDM2 (KD = 1.68 µM) and inhibiting its function. It provides some inspiration for the research into the anti-tumour target and mechanism of ganoderic acid compounds and the development of related anti-tumour candidates”.

Q4: Figure 1. The structure of GAA and compound A2 should be revised to the chemical structure of GAA and GA derivative A2. Also, it is advised to update with high-quality images using Chemdraw or similar software.

A4: Thank you for your suggestion. We have changed the size the of Figure 1 and updated a high-quality image.

Q5: Please provide more information about the overall strategy for screening and treatment with statistical tests used.

A5: The information is added in 4.3 Cell viability assay. Please let us know if you need more information.

4.3. Cell viability assay

Cell viability was determined by MTT assay. MCF-7, HepG2, SJSA-1 and HK-2 cells (6 × 103 cells /well) were seeded in 96-well plates with serum-free medium for 24 hours. Then MCF-7, HepG2, SJSA-1 cells were treated with 0.1% DMSO, 25, 50, 100 μM of GAA derivatives for 48 hours (MCF-7, HepG2, HK2) or 72 hours (SJSA-1). After 48 or 72 hours, 10 μL MTT (5 mg/mL, Beyotime) was added and incubated at 37 ℃ for 4 hours. Then 100 μL of lysate was added. After complete dissolution of the crystal, the absorbance was measured at 540 nm and expressed as the average percentage of absorbance between treated and control cells. The value for control cells was set at 100%. Cell survival rate was calculated as the ratio of the absorbance of the cells and negative control after minus the blank absorbance respectively.

Q6: At the end of the abstract and conclusion by saying "It provides some ideas for the research on the target and anti-tumor mechanism of Ganoderic acid compounds and the development of related anti-tumor candidates." In view of this, how definitive are your results, and how applicable clinically?

A6: Thank you for asking such a profound and good question. Currently, anti-tumor drugs development is still focused on broad-spectrum anti-tumor drugs. Their drug resistance and toxicity to normal cells promoted us to develop targeted anti-tumor drugs. Natural products have great potential for developing anti-tumor drug due to their low toxicity. It is of great significance for modern cancer treatment to explore their anti-tumor targets and develop more potent and effective anti-tumour targeted drugs through rational target design.

Q7: Limitations and future prospectus of the present work should be added after the conclusion section.

A7: Thank you for your very constructive suggestions. We added limitations and future perspectives of the present work at the end of the article as “Although these compounds may have weaker anti-tumour activity than other small molecule anti-tumour drugs, this study may provide insights to find the target of GAA and develop new natural product anti-cancer compounds. If we can confirm the specific targets of GAA in different diseases, we can carry out target-based rational design of GAA to greatly improve its efficacy and provide an excellent scaffold for the development of new drugs.”

Q8: I would encourage authors to provide the mechanistic approach as Graphical Abstract that would provide the thorough insights into the present investigation.

A8: Thank you for your suggestion. This is very necessary. We have added Figure 2 and relevant statement to illustrate the negative feedback regulation mode of the mdm2-p53 signaling pathway and how A2 can affect this pathway, so that readers can better understand the relationship between MDM2 and p53.

Figure 2. MDM2-p53 negative feedback regulation loop.

Q9: In Supplementary files the quality of image is better than the MS file. Please provide the high-quality images in the MS.

A9: Thank you for your suggestion. But this is the clearest MS image format we can achieve.

Round 2

Reviewer 1 Report

It appears that publication in any form at this time would be premature.

The current Western blot experimental results are insufficient to support the conclusion.

Reviewer 2 Report

I am sorry, but I have not changed my opinion on this article. I recommend to reject it.

Reviewer 3 Report

The authors have significantly improved the text compared to the previous version and because I do not know what difficulties they have at work due to the epidemic, I accept explanations. However, it is difficult for me to understand the impossibility of determining the melting point. If the authors do not have an electric apparatus, then a Kjeldahl flask with an ordinary mercury thermometer and a burner will suffice.